# Effects of Cattle Breeds and Dietary Energy Density on Intake, Growth, Carcass, and Meat Quality under Thai Feedlot Management System

**DOI:** 10.3390/ani14081186

**Published:** 2024-04-15

**Authors:** Jenwit Nusri-un, Jiraporn Kabsuk, Bhoowadol Binsulong, Kritapon Sommart

**Affiliations:** Department of Animal Science, Faculty of Agriculture, Khon Kaen University, Khon Kaen 40002, Thailand; jenwit.nu@kkumail.com (J.N.-u.); ka_jiraporn@kkumail.com (J.K.); b.bhoowadol@kkumail.com (B.B.)

**Keywords:** crossbreeding, dairy beef, energy intake, growth, carcass, meat quality

## Abstract

**Simple Summary:**

The surging global demand for beef products has spurred farmers to adopt practices enhancing both quantity and quality, particularly in selecting cattle breeds. This study addresses a knowledge gap in understanding how beef and dairy crossbreeding respond to varying energy nutrition, which is crucial for designing effective tropical feeding regimes. Over six months with thirty-six steers, the research examined the impact of different dietary metabolizable energy densities on two crossbred beef cattle—Holstein Friesian and Charolais. Despite fewer carcass yields, Holstein Friesian crossbreds demonstrate superior eating quality, reduced meat drip loss, elevated meat iron content, and lower Warner–Bratzler shear force due to increased intramuscular fat. Meat color remains consistent across beef and dairy crossbreeding. Higher dietary energy density positively influences nutrient intake and rumen fermentation but unaffected growth performance and carcass traits. The study highlights the potential of Holstein Friesian crossbred steers in intensive beef production, providing valuable insights for achieving quality meat outcomes under the Thai feedlot management system.

**Abstract:**

This study determines the effects of varying dietary metabolizable energy densities on Holstein Friesian and Charolais crossbred beef cattle in fattening phases. The research focuses on nutrient utilization, ruminal fermentation, growth performance, carcass traits, and meat quality. Thirty-six steers were used in the feeding trial that lasted for six months according to a 2 × 3 factorial arrangement in a randomized complete block design (Factor A, cattle breeds (Holstein Frisian, Charolais crossbred); B, metabolizable energy density (10.5, 11.1 and 11.8 MJ/kg DM)) with six replications. The dietary energy density had no interaction with the cattle breeds (*p* > 0.05). Despite fewer carcass yields, Holstein Friesian crossbreds indicate superior eating quality to Charolais crossbreds on drip loss, meat iron content, and Warner–Bratzler shear force due to increased intramuscular fat content (*p* < 0.05) with similar meat color (*p* > 0.05). Increased dietary energy density positively impacts nutrient and energy intake and rumen fermentation (*p* < 0.05) but did not affect growth and carcass traits (*p* > 0.05). This research suggests the potential of Holstein Friesian crossbreds for intensive beef production, providing valuable insights into optimal feeding strategies for achieving quality meat outcomes. On-farm feeding trials are needed to develop a practical and economical Thai beef feedlot management system

## 1. Introduction

The global beef products market’s increasing demand has prompted farmers to implement practices that produce quantity and quality perspectives. Consumers increasingly prioritize animal welfare, environmental sustainability, and overall eating quality [1,2]. Therefore, the selection of cattle breeds plays a vital role in determining beef production performance and meat quality. Because of the lower productivity of the Zebu breed (*Bos indicus*), most of the tropical regions have adopted the European breed (*Bos taurus*), crossbreeding to improve beef or dairy productivity. Charolais crossbreds are among the genotypes widely accepted for beef type, while Holstein Friesian is used for dairy type, each with distinct genetic and bio-resources available and backgrounds [3,4,5]. To our knowledge, understanding how these differing crossbreeding types respond to variations in energy nutrition is limited to designing tropical feeding regimes that maximize feed efficiency, growth performance, and carcass traits while ensuring desirable meat quality.

Consumer preferences are associated with eating quality attributes of tenderness, juiciness, flavor, and overall acceptability [6,7]. Intramuscular fat deposition or marbling in muscle positively correlates with meat tenderness, usually assessed by Warner–Bratzler shear force, an essential attribute concerning eating quality. The review focused on the factors influencing intramuscular fat, such as breed, sex, age, and animal nutrition [8]. The degree of intramuscular fat is associated with the dietary energy supply and body fat storage; hence, the animals’ energy intake must exceed their maintenance requirements for intramuscular fat deposition to occur in fattened beef cattle. It was previously reported that a high-energy diet increase is one way to raise energy supply because energy-dense carbohydrates can be utilized in the rumen fermentation to produce acetate and propionate, which is the primary precursor for fatty acid biosynthesis in ruminants [4,5,8]. Compared with cassava pulp, cassava chips and broken rice are suitable tropical energy-feed resources for improved rumen digestibility because they supply higher metabolizable energy [4,9]. Our previous reports also suggest that increases in the energy density of diets using cassava chips instead of cassava pulp improve beef cattle’s feed intake, digestibility, greenhouse gas emitted, and growth performance [5,10]. Still, there is a lack of information on crossbreeding differences in carcass traits and meat quality.

Dairy beef is gaining global interest among dairy producers in generating more valuable male calves, particularly its potential to improve the profitability and sustainability of the fattened crossbreeding production system [1,2,6]. Beef-on-dairy production has intensified, resulting from a reduced need for dairy heifer replacement, increasingly volatile milk prices, the potential heterosis effects in the embryo from crossbreeding, and acceptance of markets and consumers [1]. Several studies reported an improvement in the growth performance, carcass traits, and profitability of beef-on-dairy crossbreeding, suggesting that regardless of feeding strategy, crossbred beef-on-dairy cattle have more significant daily gain and are more efficient than purebred dairy steer [1,11,12]. However, a lack of literature data compares the production performance difference between Holstein Friesian and Charolais crossbreeding with Zebu cattle diets differing in energy density in tropical conditions. Therefore, this study aims to determine nutrient utilization, ruminal fermentation, growth performance, carcass traits, and meat quality in two fattened cattle breeds fed three differing dietary metabolizable energy densities in early to late fattening phases. By elucidating the responses of these breeds to varying energy densities, this study aspires to contribute valuable insights to tropical feeding regimes, optimizing feed efficiency, growth potential, and carcass traits while ensuring desirable meat quality outcomes.

## 2. Materials and Methods

The experiment was conducted at the research farm, Khon Kaen University, Khon Kaen Province, Thailand (16.46° N 102.82° E; altitude 169 m above sea level) from January 2021 to June 2021. All animal treatment and related procedures were performed according to the Khon Kaen University Animal Ethics Committee (Record No. IACC-KKU-49/62).

### 2.1. Animals, Experimental Design, and Diet

A total of thirty-six fattening beef steers, eighteen Charolais crossbred (75% Charolais × 25% native Thai; age of 24.6 ± 2 months initial body weight of 548 ± 9.90 kg) and eighteen Holstein Friesian crossbred (93.19% Holstein Friesian × 6.81% native Thai; average age of 22 ± 0.1 months initial body weight of 496 ± 17.0 kg) steers were used for the experiment. Animal was purchased at the backgrounded stage from local farmers and treated for intestinal and external parasites (1 mL/50 kg body weight; Ivermectin F, Bangkok, Thailand) and vitamins A, D3, and E (10 mL/head; Phenix, Bangkok, Thailand; vitamin A propionate, 300,000 IU/mL; vitamin D_3_ cholecalciferol, 100,000 IU/mL, vitamin E acetate, 50 mg/mL) before the start of the experiment. Each animal was housed in a pen (2.5 m × 4.5 m) with free feed and drinking water access. The pens were cleaned every morning throughout the experiment. Animals were fed ad libitum twice daily at 08.00 and 15.00 h (50:50 proportion *w*/*w*).

The individual animal was considered an experimental unit and was assigned to one of the six blocks (replicates) according to initial body weight using a 2 × 3 factorial arrangement in a randomized complete block design. Factor A was two cattle breeds (Charolais and Holstein Friesian crossbred), and factor B was three dietary metabolizable energy densities (low, medium, and high; Table 1). The feeding trial was divided into two phases (the early fattening phase lasted 59 days, and the late fattening lasted 104 days before slaughtering) after an adaptation period of 15 days. For each feeding phase, the dietary energy density was calculated to contain metabolizable energy content based on the energy content of feedstuffs according to WTSR [13] for the early fattening stage (low: 10.5, medium: 10.8, and high: 11.2 MJ ME/kg DM) and late fattening stage (low: 10.5, medium: 11.1, and high: 11.8 MJ ME/kg DM).

According to WTSR [13], the total mixed ration silage was formulated (Table 1) and produced to meet the nutrient requirements of fattening beef cattle. The total mixed ration silage was prepared in a horizontal mixer. Approximately 2000 kg of each batch of the dietary treatment mixture was mixed and packed 35 kg per plastic bag silos and, after preparation, was stored outdoors at ambient temperatures of 25 °C to 39 °C for at least seven days [4,5].

### 2.2. Sampling Collection and Laboratory Analysis

Dry matter (DM) content was determined by fan-forced oven of the samples at 105 °C to a constant weight. Subsamples were dried in a fan-forced air oven at 65 °C for 24 h, milled, and passed through a 1 mm screen and chemical analysis. Ash and ether extract (EE) was determined using the AOAC method [14] (methods 942.05 and 920.39). Dry feed samples weighing approximately 0.1 g were encapsulated in tin foil and analyzed for nitrogen (N) using a nitrogen analyzer (LECO FP828, MI, USA). Crude protein (CP) was calculated as 6.25 × N according to the methods of Etheridge et al. [15]. Fiber contents, including those of neutral detergent fiber (NDF) and acid detergent fiber (ADF), were analyzed using an Ankom200 Fibre Analyzer (Ankom Technology, Macedon, NY, USA), NDF treated with α-amylase and sodium sulfite according to the method of Mertens [16]. Non-fiber carbohydrate content was estimated using the 100 − (CP + NDF + EE + Ash) [17].

During the experiment, each morning, we collected feed-offered samples every week and pooled them for chemical analysis. The daily nutrient and energy intake were calculated as the difference between the amount of feed offered and refused.

The body weight of all cattle was recorded at the start and end of the experiment before the morning feeding. The average daily gain was calculated by the initial and final body divided by the total day of each feeding phase.

Blood and rumen fluid samples were collected from each animal three hours after the morning feeding on day 54 (for the early fattening phase) and day 103 (for the late fattening phase) after starting the experiment. Approximately 10 mL of blood was collected from the jugular vein, packed on ice, and transported to the laboratory (Accreditation No. 4138/57; Khon Kaen TLC Lab Center Co., Ltd., Khon Kaen, Thailand). Plasma metabolites (urea nitrogen, glucose, triglyceride, cholesterol, total protein, and albumin concentrations) were determined using colorimetric method test kits (Roche Diagnostics, Indianapolis, IN, USA) and an automated analyzer (COBAS INTEGRA 400 plus analyzer, Roche Diagnostics, Indianapolis, IN, USA). Two hundred milliliters of rumen fluid were taken from each animal using an esophageal–rumen stomach tube technique [5]. The ruminal fluid pH was immediately measured with a glass electrode pH meter (Eutech pH 700, Eutech Instruments Pte Ltd., Ayer Rajah Crescent, Singapore). Ruminal fluids were separated from the feed particles through four layers of gauze, and then 100 mL of rumen fluid was put into 150 mL plastic containers with 10 mL of 6 N HCL. These were collected and stored in ice buckets before being transported to the laboratory and stored at −20 °C. The concentration of NH3-N was analyzed using a using Kjeldahl nitrogen analyzer. The lactic acid and volatile fatty acid concentrations were analyzed using gas chromatography (GC2014, Shimadzu, Kyoto, Japan).

Animals were slaughtered at the end of the experiment. They were transported by truck for approximately 8 h (506 km) to a good manufacturing practice (GMP) standard commercial slaughterhouse (Prakob Beef Products Co., Ltd., Ratchaburi, Thailand). They were slaughtered after being fasted overnight but allowed free access to water. The animals were weighed, stunned, bled, skinned, eviscerated, and washed. The dressing percentages of hot carcasses were calculated. After dressing, the carcasses were chilled, aging at 4 °C to 7 d. The chilled carcass percentage was calculated similarly to the hot carcass dressing percentage.

After 7-day chilling, the rib-eye area between the 12th and 13th rib surfaces of the longissimus dorsi muscle was measured using a transfer and graph paper. In contrast, the back fat thickness was measured at 3/4 of the longissimus dorsi muscle length at the 12th rib. At the same time, approximately 1 kg of longissimus dorsi muscle samples were taken from the right side of each cattle between the 12th and 13th ribs, transported to the laboratory at 4 °C, and stored at −20 °C for further physicochemical characteristics and chemical analysis. To analyze physicochemical characteristics, the longissimus dorsi muscle samples were cut in half at a thickness of 2.5 cm steaks and then exposed to air for one hour (blooming) at room temperature. To measure surface meat color after muscle oxygenation, meat pH was using a pH meter (Consort C933 Multi-parameter analyzer, Consort bvba, Belgium), and the meat color surface was measured at three locations that were randomly selected to calculate the average values using a color reader (CR-10, Konica Minolta, Tokyo, Japan). Lightness (L*), redness (a*), and yellowness (b*) were recorded.

To determine the drip loss, 50 g meat samples were hung on a nylon cord in a plastic bag at 4 °C for 24 h. Three 2.5 cm thick steak subsamples were sealed in heat-resistant plastic bags to be boiled in a water bath (WNE 29, Memmert, Schwabach, Germany) at 80 °C until an internal temperature of 75 °C was reached. Following that, the samples, after cooking, were sliced into three subsamples parallel to the fiber orientation (width × length × height, 1 × 3 × 1 cm^3^ rectangle) before measuring Warner–Bratzler shear force with a texture analyzer (model TA. HDplusC, Stable Micro Systems, Goldalming, UK). The chemical composition of longissimus dorsi samples was determined following the standard methods of AOAC [14].

### 2.3. Statistical Analysis

All data were subjected to analysis of covariance using initial body weight as the covariate to adjust for initial weight differences among animals by the generalized linear model procedure of SAS 9.0 [18]. Nutrient intake, ruminal fermentation, plasma blood metabolites, carcass characteristics, meat quality, and chemical data were analyzed according to a 2 × 3 factorial arrangement in a randomized complete block design according to the following model:Y_ijk_ = µ + *cov* + *Blk_i_* + α_j_ + β_k_ + αβ_jk_ + ε_ijk_
where Y_ijk_ is the dependent variable, μ is the overall mean, *cov* is the covariate (initial body weight), *Blk_i_* is the effect of the block (i = 1 to 6), α_j_ is the effect of cattle breeds (j = 1 to 2), β_k_ is the effect of dietary energy density (k = 1 to 3), αβ_jk_ is the interaction effect (breeds × energy density), and ε_ijk_ is an error. Within the dietary treatments, polynomial contrasts (linear and quadratic) were performed to examine the effects of the energy density. Significance was declared at *p* ≤ 0.05.

## 3. Results

### 3.1. Feed Intake and Growth Performance

There were no significant interactions between cattle breeds and energy density for feedlot feed intake and performance measurements in each fattening phase (Table 2). Holstein Friesian cattle had a higher feed and energy intake than Charolais crossbred cattle (*p* < 0.01) in the early fattening stage. An increase in the energy density in the diet was a significantly linear increase (*p* < 0.01) in daily dry matter and metabolizable energy intake. For the late fattening stage, there was no difference in feed and energy intake between Holstein Friesian and Charolais crossbred cattle (*p* > 0.05). However, feed and energy intake increased linearly (*p* < 0.01) with increasing dietary energy density.

There were no significant interactions between cattle breeds and energy density for feedlot growth performance measurements in each fattening phase (Table 2). For the early finishing stage, Holstein Friesian crossbred cattle had a higher feed conversion ratio than Charolais (*p* < 0.01), and increasing energy density had a significant linear (*p* < 0.01) feed conversion ratio. Cattle breeds and dietary energy density did not significantly affect feedlot growth performance and feed conversion ratio (*p* > 0.05) in the late fattening phase.

### 3.2. Ruminal Fermentation Characteristics

Table 3 presents ruminal fermentation characteristics in the early and late fattening phases. No significant interactions between cattle breeds and energy density were observed for ruminal fermentation in the early fattening stage. The ruminal pH, lactic acid, and total VFAs in the Holstein Friesian crossbred were lower (*p* < 0.05) than in the Charolais crossbred. While ammonia concentration increased linearly, the pH decreased linearly (*p* < 0.05), but increasing energy density did not affect the VFA composition.

No significant interactions between cattle breeds and energy density were observed for ruminal fermentation in the late fattening stage. Charolais crossbred cattle showed higher lactic acid, propionic acid, and iso-butyric acid values than Holstein Friesian cattle (*p* < 0.05). The propionic acid increased quadratically (*p* < 0.05) with increasing dietary energy density, while acetic acid and A:P ratio decreased quadratically (*p* < 0.01) with increasing dietary energy density.

### 3.3. Plasma Metabolites

There was no significant interaction between cattle breeds and energy density for plasma metabolites in the early and late fattening phases (Table 4). For the early fattening stage, Charolais crossbred cattle had a higher blood urea nitrogen and total protein than Holstein Friesian crossbred (*p* < 0.05), and increasing the energy density had a significant quadratic effect (*p* < 0.05) in blood urea nitrogen.

Charolais crossbred cattle showed higher blood urea nitrogen, cholesterol, and total protein values for the late fattening stage than Holstein Friesian crossbred cattle (*p* < 0.05). Increasing the energy density had no significant effect on plasma metabolites.

### 3.4. Carcass Traits and Meat Quality

Carcass traits are shown in Table 5. No significant interactions exist between cattle breeds and energy density on carcass traits. The Holstein Friesian crossbred had less (*p* < 0.05) hot carcass weights, dressing percentages, fat thickness, and ribeye area than the Charolais crossbred. Dietary energy density did not significantly affect the carcass traits (*p* > 0.05).

Longissimus dorsi muscle meat quality is shown in Table 6. The interaction between breeds and energy density significantly (*p* < 0.05) affected meat pH. Dietary energy density did not significantly affect meat quality (*p* > 0.05). There was no significant difference in meat color, but meat pH, drip loss, Warner–Bratzler shear force, protein, and moisture content of the Holstein Friesian crossbred was lower (*p* < 0.05) than that of the Charolais crossbred but had higher intramuscular fat (*p* > 0.06) and iron content (*p* < 0.01).

## 4. Discussion

This study determined energy utilization, ruminal fermentation, growth performance, carcass traits, and meat quality in two fattened beef cattle breeds (Holstein Friesian vs. Charolais crossbred with native Thai beef cattle) fed three differing dietary metabolizable energy densities (low, medium, and high). Overall, the diets had no interaction with cattle breeds. When the diets have increased energy density, growth performances, carcass traits, and meat quality are unaffected, but feed and energy intake and ruminal fermentation characteristics improved. Despite Holstein Friesian crossbreds having fewer carcass yields than Charolais crossbreds, their growth performance, feed efficiency, and meat quality are similar, with greater Warner–Bratzler shear force and intramuscular fat.

### 4.1. Feed and Energy Utilization

The two-phase feeding, consisting of early and late fattening, is expected in the intensive beef production system [19,20]. The primary goal of the early fattening stage is to promote rapid growth and muscle development in young beef steers with high-energy feeds, typically rich in grains, to support efficient growth performance. These energy requirements can be achieved by providing highly digestible energy feed sources and increasing metabolizable energy to enhance animal productivity [4,5,9,21,22,23]. Dietary energy density is an essential limiting factor determining the energy supply required for feedlot cattle performance. In this study, we have formulated the dietary ingredients by substituting cassava pulp with cassava chips and broken rice to increase dietary metabolizable energy density for the late fattening stage. The results indicate that Holstein Friesian crossbred cattle showed a significantly higher feed intake than Charolais crossbred. In addition, the increasing energy density in diets is strongly associated with more excellent daily nutrients and metabolizable energy supply. Similarly, the study has reported that the increased feed intake with higher dietary energy density may be because of the decreased fiber content; thus, non-fiber carbohydrates increase substantially [5,20]. Kongphitee et al. [10] suggested that increasing non-fiber carbohydrates in the diets benefit the increase in nutrient intake and digestibility, ruminal propionate production, and energy intake, thus improving energy balance in native Thai beef cattle. They also explained that the intramuscular fat precursors, acetate, and propionate, are the glucogenic precursors mostly from ruminal fermentation, later converted to fatty acid in ruminants [4,8].

In this study, substituting cassava pulp with cassava chips and broken rice suggests an increased propionate while acetate and A:P ratio decreased with the increasing energy density in diets. Energy-dense grains can be used in the rumen to produce propionate, the primary precursor for fatty acid biosynthesis in ruminants [4,5,8]. Similarly, it was reported that the percentage of propionate increased, and the decrease in the ratio of acetate to propionate was influenced by the type of dietary degradable carbohydrate in which cassava and broken rice substituting cassava pulp were used [4,8,9,23]. These results are consistent with previous reports [21,22,24,25,26], which suggested that a higher energy density in dietary trends increases the intramuscular fat and fatty acid content in the longissimus dorsi muscle. Moreover, these results also agree with previous works [4,27] reporting that using rice grain in feed improves the digestibility of starch and protein, ruminal propionate, and marbling score.

Rumen microorganisms provide nutrients and energy for host animals by degrading and fermenting organic compounds into end products for ruminants. In this study, 10% of rice straw in the late fattening diets was used as a main forage roughage source and had high ADF and ADL content; the diet’s NDF content (range 29.4 to 43.2%) provided sufficient NDF for cattle. The minimum recommended 25 to 33% NDF was required to sufficiently stimulate chewing activity and maintain an average ruminal pH greater than 6.0 SARA risk [17]. In the present study, increasing energy density in the diets resulted in optimal blood metabolite and rumen health, with pH values ranging from 6.52 to 7.08. Additionally, despite having a very high concentrate-to-roughage ratio in the diet in this study, animals were not at risk of subacute ruminal acidosis, rumenitis, or laminitis and showed no symptoms of the illness.

### 4.2. Growth and Carcass Traits

Traditional beef-type breeds’ growth rate and carcass yield are usually better than dairy-type breeds. Several studies have been conducted to improve beef’s growth performance and profitability in dairy crossbreeding [1,2,11,12]. However, a lack of data compares the production performance difference between Holstein Friesian and Charolais crossbred with native Thai zebu cattle-fed diets differing in energy density. The result shows a similarly high growth performance of the Holstein Friesian compared with the Charolais crossbred, with the overall average daily gain in the late fattening stage (0.84 to 0.91 kg/day) compared to that in the early fattening stage (1.82 to 1.98 kg/day). The result indicated that the high rate of dry matter and energy intake may exit in the early fattening phase. Thus, a high average daily gain with a lower feed conversion ratio was observed as an influence of compensatory growth. Both cattle breeds show similar growth rates and slowdown in the late fattening stage; there may be a shift in the deposition from protein to fat tissues. The energy requirement of dairy breed animals, e.g., Holstein Friesian in this study, relative to their weight, is expected to be high because they have more active internal organs and fat depots necessary to sustain their milk production than dairy cows [1]. In addition, other studies have reported that daily weight gain decreased in the late fattening stage even though feed and energy intake is like those in the early fattening stage because they require higher maintenance energy during the late fattening stage [19,28]. Previously, no significant difference in daily gain in dairy on beef crossbred compared with purebred Holsteins [1,11] was reported. This was consistent with the results of studies on the effects of three dietary energy densities [24,25,26].

These data agreed with the recent report [1,6] that saleable meat yield of carcass traits expressed as carcass weights, dressing percentages, fat thickness, and ribeye area of Holstein Friesian was inferior to the Charolais crossbred mainly because of lower dressing percentage and muscle-to-bone ratio. However, dietary energy density under the Thai feedlot management system has similar carcass traits, suggesting that meat products could achieve a similar market premium price, carcass value, and eating quality. This also implies that crossbreeding dairy beef may represent a new intensive beef production option that aims to fatten and process more dairy cattle for beef, allowing farmers and processors to increase the profitability of both the beef and dairy industries [2,6,11].

### 4.3. Meat Quality

Consumer preferences are associated with eating quality attributes (meat color, tenderness, juiciness, and flavor) [6,7]. Post-mortem slaughter muscle pH reduced from 7.2 because glycogen is converted to lactic acid. These data suggest that the lower meat pH of the Holstein Friesian crossbred may be associated with higher glycogen deposition in the longissimus dorsi muscle than that of the Charolais crossbred. Meat pH is essential in assessing meat quality, affecting meat color, juiciness, and tenderness, with a pH of 5.4–5.8 found in standard, tender meat [29]. Therefore, the beef observed in this study can be considered normal, tender meat with pH values like those reported previously [3,25,29,30].

The bright cherry red color is a typical indicator of retail display steaks, a key factor influencing consumer acceptance and buying decisions. Major factors influence meat color, including pH, muscle type, age, animal breed, species, and feeding [12,30,31]. In this study, neither cattle breeds nor energy density affected longissimus dorsi muscle color. Foraker et al. [6] suggest that longissimus muscle from dairy cattle purebred exhibited a more significant portion of oxidative muscle fibers than dairy crossbred and conventional beef cattle. Therefore, the meat observed in this study can be similar in color between the two breeds. These results are consistent with previous reports [30], which suggested that dietary energy density and energy intake of native Thai beef cattle fed differing roughage did not affect meat color. Boonsaen et al. [3] also report similar meat color in Charolais crossbred fed differing in cassava chip or ground corn as an energy feed in total mixed ration.

Drip loss, the fluid that leaks from fresh meat, is called economic loss because the meat tissue loses weight during storage [29]. This study showed no difference in drip loss among dietary energy density treatments, consistent with Li et al. [28] and Chaosap et al. [32]. However, Lukkananukool et al. [30] reported a significantly decreased drip loss in purebred native Thai fed 1.5xmaintenance or 2.0xmaintenance energy intake level. Therefore, this study’s magnitude difference among energy feeding levels may suggest the influence on drip loss. Water holding capacity is attractive to consumers, and it could be improved by reducing drip and cooking loss through intramuscular fat [25]. These data indicated that the Holstein Friesian crossbred showed a lower drip loss than the Charolais crossbred and was associated with a more outstanding intramuscular fat content, thus changing water-holding capacity and tenderness.

Meat tenderness, usually assessed by Warner–Bratzler shear force, is essential to eating quality. These results showed no difference in shear force among dietary energy treatments. Similarly, the previous report showed that metabolizable energy density in diets has no effect on cooking loss or shear force [25]. In this study, the Warner–Bratzler shear force of the Holstein Friesian crossbred was greater, indicating a more excellent consumer preference quality than that of the Charolais crossbred.

Intramuscular fat deposition or marbling in muscle positively correlates with more excellent sensory traits, including tenderness, juiciness, flavor, and overall consumer acceptability [8]. These results showed no difference in intramuscular fat percentage among the dietary energy treatments, contrasting with previous reports [25,29]. These data agree with Kang et al. [20], who used glucogenic fattening diets, which may explain the difference in response between the lipogenic and glucogenic energy source diets reported previously [25,29]. Feeding a higher-energy feed is one strategy that encourages fat synthesis and leads to more significant fat accumulation in the carcass, which has been recommended [20,25]. However, this result indicated that the Charolais crossbred contained lean meat and lower fat and iron content, and the Holstein Friesian can explain crossbred genotypes having a more significant intramuscular fat percentage. These data suggested that crossbreeding dairy cattle could capture carcass premium in the beef supply chain, especially from a meat and carcass quality perspective.

Although the analysis of co-variance in a randomized complete block design accounted for the difference in the native Thai breed, age, initial body weight, and the limited number of cattle per treatment, this study has limitations. In particular, the production performance and meat quality results must be confirmed with a more uniform animal breed, age, and body size.

## 5. Conclusions

The influence of varied dietary metabolizable energy densities on Holstein Friesian and Charolais crossbred beef cattle of a two-phase intensive feeding was concluded. Higher energy density diets positively influenced nutrient and energy intake and rumen fermentation but unaffected growth performance and carcass traits. Despite a similar growth rate in Holstein Friesian crossbreds, they require more feed and energy intake with less carcass yield than Charolais crossbreds. The meat quality of Holstein Friesian crossbreds suggested an improved water-holding capacity and tenderness due to increased intramuscular fat when compared with Charolais crossbreds. On-farm feeding trials are needed to develop a practical and economical Thai feedlot management system.

## Figures and Tables

**Table 1 animals-14-01186-t001:** Ingredients analyzed and the chemical composition and energy content of the diets.

Item	Early Fattening Phase			Late Fattening Phase		
	Dietary Energy Density					
	Low	Medium	High	Low	Medium	High
Ingredients, % DM						
Rice straw	10	15	20	10	10	10
Cassava pulp	50	30	10	50	30	10
Cassava chip	0	15	15	0	20	20
Broken rice	0	0	15	0	0	20
Wet brewery	10	10	10	10	10	10
Palm meal	10	10	10	10	10	10
Rice bran	11	11	11	11	11	11
Soybean meal	7.5	7.5	7.5	7.5	7.5	7.5
Urea	0.5	0.5	0.5	0.5	0.5	0.5
Mineral ^1^	0.9	0.9	0.9	0.9	0.9	0.9
Premixed ^2^	0.1	0.1	0.1	0.1	0.1	0.1
Total	100	100	100	100	100	100
Chemical composition, % DM						
Dry matter	25.0	33.1	43.2	25.3	36.3	43.1
Organic matter	94.5	93.3	93.0	93.8	94.0	94.1
Crude protein	13.6	14.3	14.3	13.8	14.3	15.3
Ether extract	4.22	4.64	4.89	4.52	3.84	4.11
Neutral detergent fiber	45.3	41.6	36.3	43.2	40.8	29.4
Acid detergent fiber	21.9	21.7	19.2	24.3	20.4	15.4
Non-fiber carbohydrate ^3^	31.3	32.8	37.1	32.3	35.0	45.3
Metabolizable energy ^4^, MJ/kg DM	10.5	10.8	11.2	10.5	11.1	11.8

^1^ Minerals included Ca 93.72 g/kg; P 46.86 g/kg; Na 107.78 g/kg; S 18.55 g/kg; Mn 8.24 g/kg; Zn 7.49 g/kg; Mg 3.37 g/kg; Cu 1.17 g/kg; Co 0.15 g/kg; K 0.01 g/kg; I 0.04 g/kg; Se 0.02 g/kg (Mineral #0106410029, Dairy Farming Promotion Organization of Thailand (D.P.O.), Saraburi, Thailand). ^2^ Premix included vitamin A 5,000,000 IU/kg; vitamin D3 1,000,000 IU/kg; vitamin E 10,000 IU/kg; Cu 4 g/kg; Mg 20 g/kg; Co 0.13 g/kg; Zn 15 g/kg; I 0.75 g/kg; Se 0.38 g/kg of feed additive (Golden Mix S#0104610040, DFC Advanced Co., Ltd., Khon Kaen, Thailand). ^3^ Non-fiber carbohydrate content was estimated using the 100 − (CP + NDF + EE + Ash). ^4^ Calculated values based on the energy content of feedstuffs according to WTSR [13].

**Table 2 animals-14-01186-t002:** Effects of cattle breeds and dietary energy density on dry matter intake, metabolizable energy intake, and growth performance in Holstein Friesian (HF) and Charolais (CH) crossbred steer by fattening phases.

Item	Breed		Dietary Energy Density			SEM	*p*-Value ^1^			
	HF	CH	Low	Medium	High		B	E-L	E-Q	B × E
Dry matter intake, kg/d										
Early fattening ^2^	9.87	8.69	8.78	8.96	9.94	0.27	<0.01	<0.01	0.53	0.06
Late fattening ^3^	8.72	8.83	8.12	8.78	9.52	0.32	0.63	0.01	0.88	0.07
Dry matter intake, %BW										
Early fattening ^2^	1.81	1.43	1.52	1.58	1.70	0.04	<0.01	<0.01	0.23	0.01
Late fattening ^3^	1.35	1.27	1.23	1.30	1.39	0.04	0.56	<0.01	0.96	0.01
Energy intake, MJ ME/d										
Early fattening ^2^	107	94.2	92.2	96.7	111	2.88	<0.01	<0.01	0.42	0.04
Late fattening ^3^	97.2	98.7	85.2	98.0	112	3.60	0.58	<0.01	0.96	0.09
Body weight, kg										
Initial	496	548	525	512	537	-	-	-	-	-
Final	693	720	693	707	725	13.3	0.37	0.32	0.95	0.68
Average daily gain, kg/d										
Early fattening ^2^	1.82	1.98	1.95	2.03	1.76	0.17	0.64	0.44	0.49	0.75
Late fattening ^3^	0.84	0.91	0.82	0.85	0.99	0.13	0.79	0.43	0.77	0.82
Feed conversion ratio										
Early fattening ^2^	5.61	4.66	4.62	4.80	5.86	0.27	<0.01	<0.01	0.36	0.18
Late fattening ^3^	11.5	11.5	11.8	11.7	10.8	1.65	0.72	0.53	0.79	0.59

Abbreviations: SEM, standard error of means; ME, metabolizable energy. ^1^ B, cattle breeds, E-L, and E-Q are linear and quadratic effects of energy density, respectively; B × E is the interaction of cattle breeds and energy density. ^2^ Energy density (low = 10.5 MJ ME/kg DM, medium = 10.8 MJ ME/kg DM, high = 11.2 MJ ME/kg DM). ^3^ Energy density (low = 10.5 MJ ME/kg DM, medium = 11.1 MJ ME/kg DM, high = 11.8 MJ ME/kg DM).

**Table 3 animals-14-01186-t003:** Effects of cattle breeds and dietary energy density on ruminal fermentation characteristics in Holstein Friesian (HF) and Charolais (CH) crossbred steer by fattening phases.

Item	Breed		Dietary Energy Density			SEM	*p*-Value ^1^			
	HF	CH	Low	Medium	High		B	E-L	E-Q	B × E
Early fattening phase ^2^										
pH	6.66	7.03	7.08	6.76	6.75	0.10	0.02	0.03	0.25	0.47
Ammonia-N, mg/dL	5.31	4.93	4.38	5.25	5.69	0.34	0.60	<0.01	0.70	0.13
Lactic acid, mmol/L	0.37	1.04	0.58	0.80	0.84	0.10	<0.01	0.09	0.47	0.81
Total VFAs, mmol/L	52.4	74.8	65.9	69.5	64.9	6.44	0.01	0.98	0.65	0.85
VFA composition, %										
Acetic acid	66.7	64.2	64.0	65.4	66.6	1.07	0.01	0.17	0.64	0.23
Propionic acid	17.9	20.0	20.0	18.9	18.3	1.37	0.18	0.47	0.88	0.42
Iso-butyric acid	1.45	1.48	1.24	1.35	1.81	0.10	0.69	<0.01	0.24	0.26
Butyric acid	10.2	10.0	10.7	10.8	8.83	0.65	0.31	0.09	0.49	0.82
A:P ratio	3.82	3.42	3.42	3.57	3.82	0.24	0.05	0.36	0.91	0.47
Late fattening phase ^3^										
pH	6.65	6.57	6.58	6.54	6.71	0.10	0.23	0.35	0.62	0.95
Ammonia-N, mg/dL	5.20	4.91	4.25	6.38	4.58	0.51	0.70	0.79	0.01	0.25
Lactic acid, mmol/L	0.55	0.99	0.79	0.88	0.80	0.10	<0.01	0.82	0.33	0.79
Total VFAs, mmol/L	89.5	75.2	85.6	74.3	78.4	4.68	0.06	0.13	0.19	0.79
VFA composition, %										
Acetic acid	68.1	65.1	69.1	68.7	61.3	1.07	<0.01	<0.01	0.01	0.27
Propionic acid	16.8	19.6	16.6	16.5	22.3	1.10	<0.01	<0.01	0.03	0.90
Iso-butyric acid	1.07	1.40	1.20	1.21	1.40	0.10	0.02	0.29	0.39	0.90
Butyric acid	9.94	9.62	9.00	10.1	10.0	0.52	0.89	0.06	0.39	0.10
A:P ratio	4.26	3.49	4.26	4.27	2.89	0.23	<0.01	<0.01	0.01	0.61

Abbreviations: SEM, standard error of means; A:P ratio, acetic: propionic ratio; ME, metabolizable energy. ^1^ B, cattle breeds, E-L, and E-Q are linear and quadratic effects of energy density, respectively; B × E is the interaction of cattle breeds and energy density. ^2^ Energy density (low = 10.5 MJ ME/kg DM, medium = 10.8 MJ ME/kg DM, high = 11.2 MJ ME/kg DM). ^3^ Energy density (low = 10.5 MJ ME/kg DM, medium = 11.1 MJ ME/kg DM, high = 11.8 MJ ME/kg DM).

**Table 4 animals-14-01186-t004:** Effects of cattle breeds and dietary energy density on plasma blood metabolites in Holstein Friesian (HF) and Charolais (CH) crossbred steer by fattening phases.

Item	Breed		Dietary Energy Density			SEM	*p*-Value ^1^			
	HF	CH	Low	Medium	High		B	E-L	E-Q	B × E
Early fattening phase ^2^										
Urea-N, mg/dL	12.7	15.3	12.6	15.5	14.3	0.61	0.01	0.08	<0.01	0.80
Glucose, mg/dL	76.4	66.6	68.0	73.3	71.8	3.78	0.06	0.71	0.30	0.33
Triglyceride, mg/dL	18.3	17.9	18.2	18.7	17.3	1.67	0.56	0.73	0.42	0.49
Cholesterol, mg/dL	120	142	120	146	130	12.5	0.14	0.59	0.19	0.74
Total protein, g/dL	6.31	6.65	6.37	6.59	6.53	0.14	0.03	0.40	0.38	0.29
Albumin, g/dL	3.70	3.74	3.76	3.72	3.68	0.06	0.77	0.27	0.72	0.04
Late fattening phase ^3^										
Urea-N, mg/dL	10.7	13.8	11.6	13.0	12.5	0.90	<0.01	0.57	0.39	0.90
Glucose, mg/dL	70.3	71.7	70.7	71.9	70.5	1.63	0.35	0.87	0.59	0.44
Triglyceride, mg/dL	22.5	27.7	24.6	27.5	23.9	3.00	0.11	0.94	0.15	0.66
Cholesterol, mg/dL	87.0	146	101	131	126	12.7	<0.01	0.19	0.23	0.48
Total protein, g/dL	7.08	7.41	7.44	7.09	7.25	0.12	0.02	0.54	0.68	0.12
Albumin, g/dL	3.76	3.88	3.87	3.88	3.73	0.06	0.07	0.19	0.16	0.09

Abbreviations: SEM, standard error of means; ME, metabolizable energy. ^1^ B, cattle breeds, E-L, and E-Q are linear and quadratic effects of energy density, respectively; B × E is the interaction of cattle breeds and energy density. ^2^ Energy density (low = 10.5 MJ ME/kg DM, medium = 10.8 MJ ME/kg DM, high = 11.2 MJ ME/kg DM). ^3^ Energy density (low = 10.5 MJ ME/kg DM, medium = 11.1 MJ ME/kg DM, high = 11.8 MJ ME/kg DM).

**Table 5 animals-14-01186-t005:** Effects of cattle breeds and dietary energy density on carcass traits in Holstein Friesian (HF) and Charolais (CH) crossbred fattened steer.

Item	Breed		Dietary Energy Density ^2^			SEM	*p*-Value ^1^			
	HF	CH	Low	Medium	High		B	E-L	E-Q	B × E
Carcass traits										
Hot carcass weight, kg	342	382	360	365	367	7.94	<0.01	0.52	0.88	0.74
Dressing percentage, %	55.6	57.5	56.2	56.5	57.3	0.64	0.01	0.16	0.44	0.31
12th rib fat thickness, cm	8.87	17.7	15.8	11.6	13.6	1.99	<0.01	0.42	0.37	0.20
Ribeye area, cm^2^	53.1	74.1	65.5	66.3	61.8	3.15	<0.01	0.38	0.54	0.59

Abbreviations: SEM, standard error of means; ME, metabolizable energy. ^1^ B, cattle breeds, E-L, and E-Q are linear and quadratic effects of energy density, respectively; B × E is the interaction of cattle breeds and energy density. ^2^ Energy density (low = 10.5 MJ ME/kg DM, medium = 11.1 MJ ME/kg DM, high = 11.8 MJ ME/kg DM).

**Table 6 animals-14-01186-t006:** Effects of cattle breeds and dietary energy density on longissimus dorsi muscle meat quality and chemical composition in Holstein Friesian (HF) and Charolais (CH) crossbred fattened steer.

Item	Breed		Dietary Energy Density ^2^			SEM	*p*-Value ^1^			
	HF	CH	Low	Medium	High		B	E-L	E-Q	B × E
pH	5.40	5.49	5.42	5.49	5.45	0.04	0.03	0.62	0.14	0.01
Color										
Lightness (L*)	33.5	33.6	33.9	33.5	33.2	1.05	0.88	0.57	0.95	0.99
Redness (a*)	22.5	21.3	21.3	22.8	21.5	0.83	0.31	0.91	0.15	0.76
Yellowness (b*)	14.5	14.2	14.1	14.9	13.9	0.44	0.51	0.81	0.10	0.71
Drip loss, %	5.36	9.81	7.08	7.96	8.33	0.73	<0.01	0.23	0.78	0.99
Warner–Bratzler shear force, kg	3.42	4.49	3.84	3.93	4.10	0.27	<0.01	0.59	0.86	0.89
Chemical composition (%)										
Moisture	69.1	70.2	69.9	69.8	69.4	0.57	0.04	0.37	0.79	0.07
Crude protein	21.6	22.6	21.6	23.0	21.7	0.46	0.06	0.90	0.03	0.94
Intramuscular fat	7.14	5.30	6.37	5.79	6.34	0.75	0.06	0.83	0.48	0.17
Ash	1.13	1.08	1.05	1.04	1.22	0.07	0.67	0.08	0.25	0.18
Iron, mg Fe/kg	47.8	37.7	44.1	40.1	43.5	3.52	0.01	0.94	0.19	0.75

Abbreviations: SEM, standard error of means; ME, metabolizable energy. ^1^ B, cattle breeds, E-L, and E-Q are linear and quadratic effects of energy density, respectively; B × E is the interaction of breeds and energy density. ^2^ Energy density (low = 10.5 MJ ME/kg DM, medium = 11.1 MJ ME/kg DM, high = 11.8 MJ ME/kg DM).

## Data Availability

Data are available from the corresponding author upon reasonable request.

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
