# Peer review of "Effects of Cattle Breeds and Dietary Energy Density on Intake, Growth, Carcass, and Meat Quality under Thai Feedlot Management System"

_animals, 2024, doi:10.3390/ani14081186_

Round 1

Reviewer 1 Report (Previous Reviewer 3)

Comments and Suggestions for Authors

I have read the revised manuscript and appreciate the authors' consideration of my previous suggestions. Authors have covered all of my observations in an acceptable manner, in such a way that I have no further observations 

Author Response

Response: We thank you for your insightful review and valuable comments, which helped us improve the manuscript.

Reviewer 2 Report (Previous Reviewer 2)

Comments and Suggestions for Authors

The authors examined the effects of cattle genotype and dietary energy density on nutrient utilization, growth performance, carcass characteristics, and meat quality. The subject of this study is suitable for the “Animals” journal. The authors have made all the previous improvements in the final article. The study is well-designed and presented, and the manuscript can be accepted.

Also;

-The results are convincing and supported by the discussion.

-The conclusions are consistent with the evidence and arguments.

-The topic and references are appropriate.

-Tables and figures are presented clearly and understandably.

Author Response

Response: We thank you for your insightful review and valuable comments, which helped us improve the manuscript.

Reviewer 3 Report (New Reviewer)

Comments and Suggestions for Authors

The manuscript attempts to evaluate the performance of different cattle breeds (including crossbreeds) under different energy nutrition conditions. The results showed that dairy beef (crossbreeds) performs quite well in tropical environments under different nutrition conditions, except for carcass yields. This study provides new insights into the nutrition management of dairy beef cattle in tropical environments such as Thailand.  Thus, the manuscript deserves publication. However, two issues need to be addressed in the revision.

1.  The authors treated different cattle breeds including crossbreeds (Dairy beef) as the genotypes. This could bring confusion because the genotypes are generally genetic information, for instance, AA, AB, and BB. Without SNP data, I strongly suggest the authors just compare the performance of different cattle breeds (including crossbreeds - dairy beef) under different nutrition conditions in tropical regions. Alternatively, the authors should give a clear definition of the term genotype in this study. 

2. Given that the same panel of cattle was analyzed in different energy nutrition conditions, this can cause multiple replications, resulting in false positives. The significance level p<0.05 should be corrected, for example, simply using p<0.05/n.

3. Cross-page tables should be avoided.

Author Response

Reviewer 3 Comments and Suggestions for Authors

The manuscript attempts to evaluate the performance of different cattle breeds (including crossbreeds) under different energy nutrition conditions. The results showed that dairy beef (crossbreeds) performs quite well in tropical environments under different nutrition conditions, except for carcass yields. This study provides new insights into the nutrition management of dairy beef cattle in tropical environments such as Thailand.  Thus, the manuscript deserves publication. However, two issues need to be addressed in the revision.

Response: We thank you for your insightful review and valuable comments, which helped us improve the manuscript.

Point 1. The authors treated different cattle breeds including crossbreeds (Dairy beef) as the genotypes. This could bring confusion because the genotypes are generally genetic information, for instance, AA, AB, and BB. Without SNP data, I strongly suggest the authors just compare the performance of different cattle breeds (including crossbreeds - dairy beef) under different nutrition conditions in tropical regions. Alternatively, the authors should give a clear definition of the term genotype in this study. 

Response 1: Thank you for your recommendation. To improve clarity, we changed “genotype” to “cattle breeds” in the title and all the manuscripts.

Point 2. Given that the same panel of cattle was analyzed in different energy nutrition conditions, this can cause multiple replications, resulting in false positives. The significance level p<0.05 should be corrected, for example, simply using p<0.05/n.

                Response 1: Based on the polynomial orthogonal contrast of dietary energy density effect, we assumed type I error or Familywise error rate (FWER) to be within 5%, the same as the breed main factor F-TEST models because we did not conduct a Post Hoc Test of ANOVA multiple comparison procedures, such as Fisher’s LSD, Tukey’s test, Bonferroni correction, in this study. Therefore, we indicated the significant declared at p<0.05, and the highly significant declared at p<0.01.

Point 3. Cross-page tables should be avoided.

Response 3: Thank you for your observation. We corrected it in all Tables.

Round 2

Reviewer 3 Report (New Reviewer)

Comments and Suggestions for Authors

The authors addressed my concerns and thus I recommend accepting it for publication.

This manuscript is a resubmission of an earlier submission. The following is a list of the peer review reports and author responses from that submission.

Round 1

Reviewer 1 Report

Comments and Suggestions for Authors

Within the simple summary, authors indicate that this study addresses the knowledge gap in understanding how Bos Taurau and Bos indicus crossbred genotypes…; however, the project is comparing beef versus dairy breeds. 

The experiment is designed correctly, and the statistical analysis was done according to the design; however, the lack of power with animal numbers minimizes the possibility of properly testing the hypothesis.

Line 14, authors indicate that over six months; however, in the material and methods. The authors only indicate the project to be January 2021 to April 2021.  This is not six months.

Line 53: ( hanging out, should be removed.

Line 90: The fattening phase was 163 day; however, the Materials and Methods stated that the research was conducted January 2021 to April 2021. 

The authors are comparing crossbred animals; however, the about of the Native Thai portion is different between the Charolais and Holstein Friesian animals.  Hence this limits the authors ability to clearly define if the differences found were due to the majority breed or the percentage of Native Thai.  Additionally, the age of the animals at the start of the project were slightly different, so could the difference found in meat characteristics be due to physiological age of the animals or genotype?

Line 119: need a period at sentence.

Table 1: Footnotes normally do not have ) following the number. You indicated that there is a 3 footnote for Non-fiber carbohydrate; however, there is no information included at the bottom of the table.

Line 168: four should be numerical 4.

Line 177: four should be numerical 4.

Line 179: space before the degree symbols. 

Line 190 – 194: Defining the formula needs to be clearer.  Readers should be able to mix the section of the formula with each definition. Explanation did not match the formula.

Results: There are numerous discrepancies between the text and data included in the tables.  I’m not sure that I will get all of them. 

                Line 204: Text has “there was no difference in the nutrient intake between Holstein-Friesian and Charolais cattle” however, on table 2 there is a p value <0.01 for genotype. 

                Line 206 – 209: Text does not match table. 

All Tables: remove ) after footnote indicates on table and within footnotes.  You have D x E in the footnotes; however, the table is G x E. 

Line 219 - 223: Text does not match table 3. 

Line 222 – 223: You have P as a capital letter; however, other areas you use a small p for p value. Be consistent.

Line 228- 231: Text does not match results in table 3.

Line 246: is it linear or quadratic? Based on the information in the table it is quadratic.

Line 260-261: Authors had “Genotypes and dietary energy density did not significantly affect feedlot growth and performances (p>0.05)”; however, initial body weight, final body weight and feed conversion, were different.  I understand that average daily gain was not different; however, final body weight does have an impact on feedlot performance.  This was created because the calves were of different weights coming into the project.  A flaw of the project.

Line 270: no – between 6 and no

Line 275: where is table 7?

The conclusion in the discussion does not match the finding of the results.  It happens that the authors are trying to make the research say what they want instead of clearly reporting what the results showed.  The discussion and conclusion need to match the findings. 

Comments on the Quality of English Language

n/a

Reviewer 2 Report

Comments and Suggestions for Authors

The authors examined the effects of cattle genotype and dietary energy density on nutrient utilization, growth performance, carcass characteristics, and meat quality. The subject of this study is suitable for the “Animals” journal. The authors suggest that Holstein-Friesian crossbreds may be more suitable for intensive beef production systems in the tropics, and the authors claim that the study provides valuable insight into optimal feeding strategies to achieve quality meat results under varying nutritional energy conditions in the tropics. The study is well designed and presented, but I offer a few corrections to increase the scientific value of the manuscript. After the authors address these corrections, the manuscript can be accepted.

-Information should be given about the animals' growing and physiological conditions before they were fattened. For example, birth weights, weaning ages, and weights, dam ages, etc.

-Diets need to verify whether the metabolic energy value is a calculated value or a measured value. If it is a calculated value, indicate below the table where the metabolic energy is the calculated value and add the formula.

-More detailed information should be given about how plasma metabolites were determined.

-The statistical analysis section should add a power analysis showing that the number of animals is sufficient. Otherwise, the reliability of the results obtained may be questioned.

-The results have been well presented, and the discussion has supported the results, but the conclusion section should be improved, especially the effect of cattle genotype and dietary energy density on animal health, welfare, and yield.

Also;

-The results are convincing and supported by the discussion.

-The conclusions are consistent with the evidence and arguments.

-The topic and references are appropriate.

-Tables and figures are presented clearly and understandably.

Reviewer 3 Report

Comments and Suggestions for Authors

General

Respected authors

All opinion was emitted with all respect to the efforts of the authors for the preparation of the experiment and its report

The purpose of this study was to evaluate the effects of the effects of two genotypes and dietary energy density on nutrient utilization, growth performance, carcass traits and meat quality in cattle fattening under Thai feedlot management system. Variables measured were growth performance, rumen fermentation characteristics, blood metabolites and some carcass traits. Justification of the experiment is clear and methodology and statistical procedures were appropriate to fully reach the objective raised. In general, the experiment was carried out in an acceptable manner. However, there exists two observations which they must be attended to. One of them is regarding the differences on initial BW between genders. It is well known that initial weight affects DMI and rate weight gain among other productive variables in feedlot cattle. In this report, Holstein cattle were 9.5% lighter than Charolais cattle. This is not a minor case! For that, the effects of gender can be confused with the effect of initial weight. For example, differences in final weight, hot carcass weight and ribeye area were mainly by initial weight not for gender. This limiting must be mentioned by the authors in the manuscript.

Another observation is that the energy density of diets used seems somewhat low, according to the majority of feedlot industry (13.5 to 14.5 Mj/kg diet). Thus, compared to current feedlot systems, the energy density of the diet used in the present experiment is in the range of "low-to-moderate". Furthermore, the high energy diet used in the present experiment represents only 85% of the energy contained in a conventional finishing diet. I think opportune that Title include “Under Thai Feedlot Management System “

Specific

Title: include “Under Thai management feedlot system” (Hint): Effects of Genotypes and Dietary Energy Density on Nutrient Utilization, Growth Performance, Carcass Traits and Meat Quality in Cattle fattening under Thai Feedlot Management System

L100: Specify IU of each vitamin instead mL of the product used

L101: were individually housed in pens

L103: Please specify the proportion of the total amount of feed offered. For example: Animals were fed ad libitum twice daily at 08.00 and 15.00 h in a proportion approximately of 50:50 of the total fed intake (or 40:60, 30:70) please specify.

L113: What does it mean by “The total mixed ration silage “

L137: Mertens. Correct it please

L149: After reading ruminal pH. How did you conserve the ruminal fluid for VFA analysis? Describe it please

L150: indicate model pf pH meter

L151: Specify model of nitrogen analyzer (Leco FP528, Lec, St. Joseph, MI) and mention the method used

L148: Please redact more clear the statement “Two hundred milliliters of rumen fluid were taken from each animal using esophageal tube connected to a rumen stomach tube pump” and include description (laboratory, commercial house, etc.) of the material and equipment used to obtain ruminal samples

L157: Indicate the distance (in km) from experimental facilities to slaughterhouse

L158: According to the standard practices? or according to rules of ethical procedures for handling and sacrifice of animals?

L181: Use superscript to denote “cubic”

L195 Results

As mentioned previously, exists an observation regarding the differences on initial BW between genders. It is well known that initial weight affects DMI and rate weight gain among other productive variables in feedlot cattle. In this report, Holstein cattle were 9.5% lighter than Charolais cattle. This is not a minor case! For that, the effects of gender can be confused with the effect of initial weight. For example, differences in final weight, hot carcass weight and ribeye area were mainly by initial weight not for gender. This limiting must be mentioned by the authors in the manuscript.

L210: Table 2. Please include here the analysis of relative DM consumption (i.e. DMI, g/kg body weight or DMI g/kg BW0.75).

L262: Table 5

a) Please check the weight data, according to initial weight, average daily gain, and duration of phases the final weight did not coincide.

b) The average daily weight gains not coincide with energy intake (this can be confirmed by performing energy derivations calculus-observed-to-expected diet NE). For example, in the first phase, the ADG observed to HF (1.82 kg/d) with a DMI of 9.87 kg, is possible only if diet contained 18.3 MJ ME/kg (2.83 Mcal NEm/kg)! Or if HF consumed 16.83 kg/d of diet contained 11.8 MJ ME/kg. This discrepancy it is difficult to explain. However, in the first phase an effect of compensatory growth can be a factor to confuse the “real’ response to diets energy density on growth-performance. The other factor could be the type of diet used. It is not rare that when high-moisture diets are offered there is a greater margin of error in accurately estimating DM intake (it is generally underestimated) which leads to estimation errors. I encourage the authors that this be mentioned in the manuscript. 

L308: Redact in third person. Avoid “our results” Start the sentence as: The results indicate

L308: Respected authors, at comparable weights, the greater DM intake (approximately 8-10%) of Holsteins breeds is expected because the Holsteins have a greater maintenance energy requirements and visceral organ mass than beef breeds (Zinn et al., 2004). It is important to note that even when Holsteins were lighter than Charolais cattle, Holstein consumed more feed and thus energy when expressed both as absolute (kg/d) and as relative (kg/ kg body weight). Please, note that in the early fattening phase Holstein consumed 12% more of feed than Charolais cattle, but at late fattening phase there is no difference in DM intake. This can be explained because 1) the growth impetus of Holstein at the early stage (associated with the low energy content in diet), and 2) in late fattening this effect (greater DM intake for Holstein) was masked because of the differences of weight between genders. Here the importance of the relative expression of DM intake must be included. Please considers this points in your discussions.

L322: “increasing energy density in the diets resulted in optimal blood metabolite and rumen health” What is the basis for this statement? Data in Tables did not support it. Correct this.

323: “despite having a very high concentrate-to-roughage ratio in the diet, animals were not at risk of subacute ruminal acidosis" What is the basis for this statement? According to Table 1, “high-energy” diet contained 29% NDF. Based on diet composition and feed intake, the expected ruminal pH is in agreement with this expectation (6.63 vs 6.74; NASEM, 2016). As mentioned above, the energy density of the diet used in the present experiment is in the range of "low-to-moderate". High energy diets (for feedlot cattle systems) are considered from 13.5 to 14.5 MJ ME/kg diet, Thus the high energy diet used in the present experiment represents only 85% of the energy contained in a conventional finishing diet. Please, rewrite the statement in congruence.

L328-342: Respected authors, remove from discussion this. None of these variables were measured here. These statements are simple filler.

L363-365: Specify “under Thai feedlot management”

Conclusions

L424: Use third person. (i.e.) It was concluded that…

L426: under Thai feedlot management system